# FACT-CHECKING WITH LARGE LANGUAGE MODELS VIA PROBABILISTIC CERTAINTY AND CONSISTENCY

## ABSTRACT

Large language models (LLMs) are increasingly used in applications requiring factual accuracy, yet their outputs often contain hallucinated responses. While fact-checking can mitigate these errors, existing methods typically retrieve external evidence indiscriminately, overlooking the model's internal knowledge and potentially introducing irrelevant noise. Moreover, current systems lack targeted mechanisms to resolve specific uncertainties in the model's reasoning. Inspired by how humans fact-check, we argue that LLMs should adaptively decide whether to rely on internal knowledge or initiate retrieval based on their confidence in a given claim. We introduce Probabilistic Certainty and Consistency (PCC), a framework that estimates factual confidence by jointly modeling an LLM's probabilistic certainty and reasoning consistency. These confidence signals enable an adaptive verification strategy: the model answers directly when confident, triggers targeted retrieval when uncertain or inconsistent, and escalates to deep search when ambiguity is high. Our confidence-guided routing mechanism ensures that retrieval is invoked only when necessary, improving both efficiency and reliability. Extensive experiments across three challenging benchmarks show that PCC achieves better uncertainty quantification than verbalized confidence and consistently outperforms strong LLM-based fact-checking baselines. Furthermore, we demonstrate that PCC generalizes well across various LLMs.

## 1 INTRODUCTION

Despite remarkable progress in recent years, factuality remains a key challenge (Augenstein et al., 2024). Large language models (LLMs) remain prone to *hallucination* (Huang et al., 2025a), often producing outputs with factual inaccuracies. Such errors can arise even when responses appear plausible and well-reasoned, making them difficult to detect without explicit verification. This limitation undermines the reliability of LLMs in real-world applications (Huang et al., 2025b). To address this, *automated fact-checking* (Guo et al., 2022) offers a promising safeguard by verifying generated claims against trusted evidence sources.

With the advent of powerful LLMs, fact-checking systems have increasingly shifted from traditional supervised methods to LLM-based approaches (Wang et al., 2024; Li et al., 2024; Wei et al., 2024; Xie et al., 2025). Most existing LLM fact-checkers follow a *retrieval-then-verification* paradigm: given a claim, the system first retrieves candidate documents from a knowledge base or the web and then determines whether the claim is supported, refuted, or unverifiable. While effective, this pipeline treats retrieval as mandatory, even when the LLM could reach a correct judgment using only its parametric knowledge (Luo et al., 2023), resulting in unnecessary search API calls. Moreover, a uniform retrieval strategy contrasts with human information-seeking behavior: for common-sense knowledge, people often rely on memory alone, whereas for specialized or domain-specific claims, they deliberately conduct deeper searches.

Since LLMs are pretrained on vast corpora, they encode substantial amounts of world knowledge in their parameters (Yu et al., 2023), enabling them to verify many claims without external retrieval. Analogous to how human fact-checkers recall known facts before consulting outside sources, LLMs could benefit from estimating their own **factual confidence** (Mahaut et al., 2024) to decide whether retrieval is necessary (Chuang et al., 2024; 2025; Farquhar et al., 2024; Tao et al., 2024). Prior work (Xie et al., 2025) has explored *verbal confidence* as such a signal, where the LLM self-reports

its confidence in relying on parametric knowledge to verify a claim and then chooses whether to retrieve. However, verbal confidence has clear limitations: it depends heavily on model calibration, which varies across LLMs (Kumar et al., 2024), is sensitive to prompt design, and often lacks robustness across tasks and domains (Yang et al., 2024).

To provide a more reliable and generalizable measure of factual confidence, we propose **Probabilistic Certainty and Consistency (PCC)**, a framework that evaluates an LLM's confidence along *two complementary dimensions*. Just as an LLM outputs both a verdict and a supporting rationale, its confidence should reflect both decisiveness and stability (Becker & Soatto, 2024). The first dimension, *internal certainty*, quantifies the model's confidence in its verdict token via the log-probability margin, capturing how strongly the output distribution favors one label over the other. The second dimension, *reasoning consistency*, assesses the stability of the model's explanations by comparing rationales generated under opposing assumptions (i.e., assuming the claim is true vs. false). A Natural Language Inference (NLI) model scores the degree of contradiction between these rationales, and consistency is defined as the complement of this score. Internal certainty thus reflects the decisiveness of the model's verdict, while reasoning consistency captures the coherence of its reasoning under adversarial framing. Together, these dimensions offer complementary signals: a model may be highly certain yet inconsistent, indicating overconfidence, or consistent yet uncertain, suggesting incomplete knowledge. By integrating both signals, PCC provides a robust and interpretable estimate of factual confidence, more generalizable across models.

To operationalize PCC for fact verification, we use its confidence signals as a decision router to construct an *adaptive verification pipeline* that tailors the verification strategy to the model's confidence profile. As illustrated in Figure 1, the pipeline partitions claims into four quadrants based on internal certainty and reasoning consistency. When both signals are high, indicating strong factual confidence, the system issues a direct answer without retrieval. When certainty is high but consistency is low, suggesting overconfident hallucination, the system triggers a *targeted search* using queries derived from the most contradictory rationale pairs, focusing retrieval on the disputed knowledge. When certainty is low but consistency is high, typically reflecting incomplete knowledge, the model is prompted to *reflect* on missing information before retrieving supporting evidence (Zhang et al., 2024). Finally, when both signals are low,

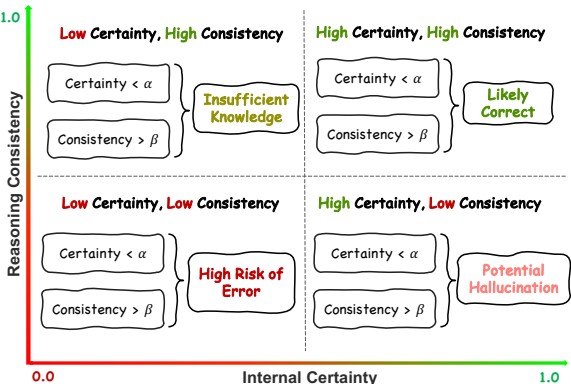

Figure 1: Illustration of how **Probabilistic Certainty and Consistency (PCC)** estimates an LLM's factual confidence along two dimensions: internal certainty and reasoning consistency. Each claim is assigned to a distinct quadrant, and a tailored verification strategy is used accordingly.

the system initiates a *deep search* procedure (Xi et al., 2025) that iteratively retrieves and assesses evidence, interleaved with self-reflection on whether the current information suffices to verify the claim. This adaptive design avoids unnecessary retrieval in confident cases while allocating more effort to ambiguous or high-risk instances.

We evaluate the calibration of PCC against standard verbal confidence and show that PCC consistently delivers superior performance, with lower Expected Calibration Error (ECE) (Guo et al., 2017) across diverse proprietary and open-source LLM families on three challenging fact-checking datasets. We further demonstrate that a PCC-guided fact-checker outperforms existing LLM-based approaches, achieving improvements of up to 15.2% on false claims, which are typically the most difficult to verify. Our experiments also highlight the strong generalization of PCC-guided verification across both proprietary and open-source models. Finally, ablation studies confirm the effectiveness of PCC as a factual confidence signal and show how it guides more targeted and efficient retrieval. In summary, our contributions are as follows:

- We propose **Probabilistic Certainty and Consistency (PCC)**, a novel framework for LLM factual confidence estimation that jointly models internal certainty and reasoning consistency, achieving more reliable calibration than verbal confidence across models and datasets.

- We design an adaptive fact-checking pipeline that leverages PCC to dynamically choose among direct answering, targeted retrieval, reflection-guided retrieval, and deep search.

- We conduct extensive experiments on three challenging fact-checking benchmarks, demonstrating that PCC-guided verification improves factual accuracy and provides new insights into the relationship between confidence signals and factual correctness.

## 2 FACTUAL CONFIDENCE ESTIMATION: CERTAINTY AND CONSISTENCY

We introduce **Probabilistic Certainty and Consistency** (PCC), a framework for estimating an LLM's *factual confidence* through two complementary dimensions. This approach is inspired by how a human fact-checker makes trustworthy judgments, not only by issuing a decisive verdict, but also by providing a coherent rationale that holds up under counterfactual scrutiny. As illustrated in Figure 2, given a claim $c$, our objective is to estimate the probability that the model's verdict on $c$ is factually correct, *without* relying on external retrieval. PCC decomposes this estimate into two signals: (1) *internal certainty*, which measures the probabilistic confidence in the model's chosen verdict, and (2) *reasoning consistency*, which assesses the logical stability of its explanations when the model is prompted to reason under opposing assumptions.

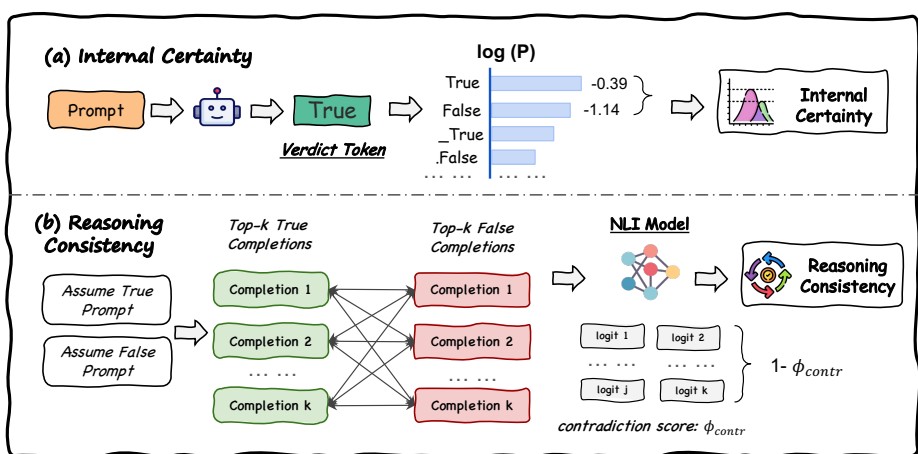

Figure 2: Illustration of **Probabilistic Certainty and Consistency (PCC)** framework. *Internal certainty* reflects the model's probabilistic confidence in its predicted verdict, while *reasoning consistency* quantifies the logical coherence of its explanations across counterfactual reasoning.

### 2.1 INTERNAL CERTAINTY

To estimate how confident the model is in its verdict, we examine the probability distribution over the next-token output when the LLM is prompted to verify a claim (see Appendix F). Given a claim $c$, let the model produce a probability distribution $P(\cdot \mid c)$ with corresponding logits $\{\ell_w\}_{w \in \mathcal{V}}$ over the vocabulary $\mathcal{V}$. We define two disjoint indicator sets over tokens: $\mathcal{T}$ for those indicating a "True" verdict, and $\mathcal{F}$ for "False." The class-level probabilities are then computed as:

$$p_{\mathrm{T}} = \sum_{w \in \mathcal{T}} P(w \mid c), \qquad p_{\mathrm{F}} = \sum_{w \in \mathcal{F}} P(w \mid c).$$

Let $t_{(1)}$ and $t_{(2)}$ denote the top two tokens ranked by probability, and let $g(w) \in \{\texttt{True}, \texttt{False}\}$ map each token to its associated verdict class. We define the *internal certainty* score $\tau(c)$ as:

$$\tau(c) = \begin{cases} 1, & \text{if } g\big(t_{(1)}\big) = g\big(t_{(2)}\big) \in \{\texttt{True}, \texttt{False}\}, \\ \big| p_{\mathrm{T}} - p_{\mathrm{F}} \big|, & \text{otherwise.} \end{cases}$$

This formulation assigns maximal confidence ($\tau(c) = 1$) when the top two predicted tokens agree on the same verdict class, indicating a decisive and locally stable preference in the model's output distribution. In cases of disagreement, we use the absolute difference between the aggregated class probabilities to quantify uncertainty, yielding a continuous score in $[0, 1]$. Unlike verbalized confidence estimates, this method relies directly on token-level probabilities, making $\tau(c)$ less sensitive to prompt variation and more robust across different model architectures and decoding settings.

## 2.2 REASONING CONSISTENCY

While internal certainty captures output-level decisiveness, it does not evaluate whether the model's reasoning remains coherent when subjected to adversarial framing. A reliable fact-checker *should not be easily swayed by counterfactual assumptions*. To probe this, we elicit two sets of rationales:

$$R^+(c) = \{r_1^+, \ldots, r_K^+\}, \qquad R^-(c) = \{r_1^-, \ldots, r_K^-\},$$

where $R^+(c)$ contains explanations generated under the assumption that the claim $c$ is true, and $R^-(c)$ contains explanations generated under the assumption that $c$ is false. The set of prompts are listed in Appendix F.

Consider the claim *"The capital of Australia is Sydney."* A knowledgeable model may generate a true-side rationale, such as *"This is incorrect; the capital of Australia is Canberra, not Sydney."* and a false-side rationale such as *"Canberra is the official capital, and Sydney, though prominent, is not the capital."* Both rationales reinforce the same factual core, yielding high consistency. By contrast, an uncertain model may hedge, producing rationales that overlap without clear logical opposition, such as *"Sydney is often mistaken for the capital due to its prominence"* (assume true) and *"Canberra is the capital, though Sydney is culturally significant"* (assume false), resulting in weaker consistency.

Formally, for each $(r_i^+, r_j^-)$ pair, we compute a contradiction probability $\phi_{\text{contr}}(u, v) \in [0, 1]$ using a Natural Language Inference (NLI) model. The mean cross-assumption contradiction is

$$\bar{\phi}_{\text{contr}}(c) = \frac{1}{K^2} \sum_{i=1}^{K} \sum_{j=1}^{K} \frac{\phi_{\text{contr}}(r_i^+, r_j^-) + \phi_{\text{contr}}(r_j^-, r_i^+)}{2}.$$

We then define *reasoning consistency* $\gamma$ as:

$$\gamma(c) = 1 - \bar{\phi}_{\text{contr}}(c),$$

so that $\gamma(c) \in [0, 1]$, with higher values indicating greater alignment between rationales across opposing assumptions.

## 2.3 EVALUATION OF FACTUAL CONFIDENCE QUANTIFICATION

We assess PCC on three fact-checking benchmarks, SCIFACT (Wadden et al., 2020), HOVER (Jiang et al., 2020), and FELMWK (Zhao et al., 2023). We use both closed-source models (GPT-4o, GPT-4o-mini, Gemini-2.5-Pro, Gemini-2.5-Flash) and open-source models (Mistral-7B-Instruct). Calibration is measured by *Expected Calibration Error* (ECE) (Guo et al., 2017), comparing PCC to three baselines: verbal confidence, internal certainty, and reasoning consistency.

Across all datasets and models, PCC consistently yields the lowest ECE (Figure 3). On SCIFACT, for instance, it reduces ECE from 23.53 (verbal) to 12.91 with Gemini-2.5-Pro. On HOVER, PCC lowers ECE from 39.37 (verbal) to 20.76 with GPT-4o-mini. Comparable improvements are observed on FELMWK, where ECE drops from 34.41 to 23.95.

These results highlight three key findings: (i) verbal confidence is systematically overconfident and poorly aligned with accuracy, (ii) internal certainty and reasoning consistency each capture only partial signals, and (iii) harmonically combining them yields consistently better calibration. PCC thus provides a robust and generalizable estimate of factual confidence, improving reliability across datasets, model families, and distributional settings. Further results, including score distribution, AUROC, and error analyses, are included in Appendix E.2, Appendix E.3, and Appendix E.4.

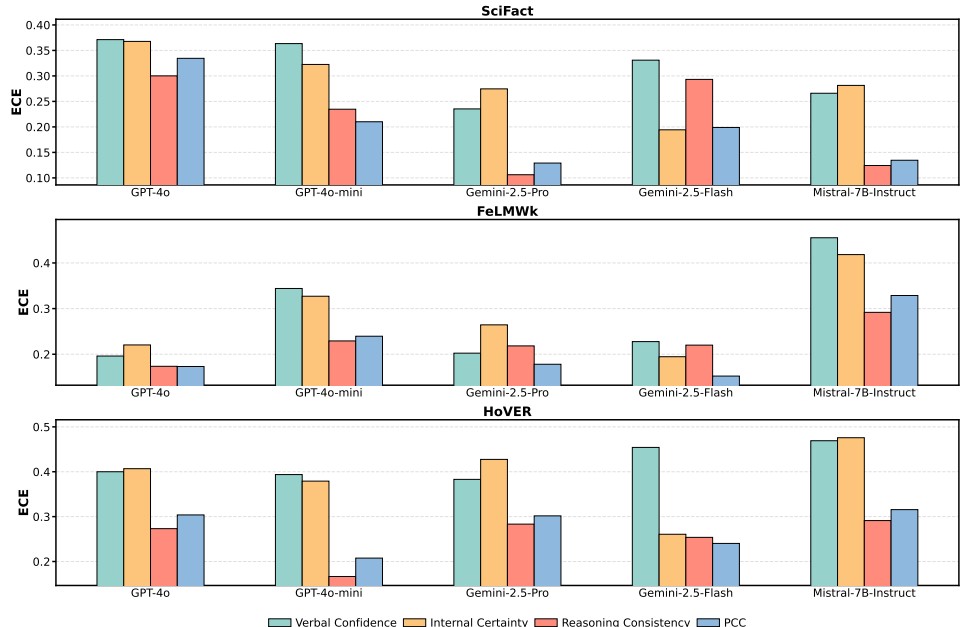

Figure 3: Expected Calibration Error (ECE) of PCC versus verbal confidence on SCIFACT, FELMWK, and HOVER. Lower ECE indicates better-calibrated factual confidence. PCC consistently achieves superior calibration across all datasets and model families.

## 3 CONFIDENCE-GUIDED FACT-CHECKING

Our confidence-guided fact-checking pipeline leverages the certainty and consistency signals $(\tau(c), \gamma(c))$ from PCC to adaptively select tailored verification strategies for each claim $c$. Here, $\tau(c)$ denotes the model's *internal certainty*, derived from log-probability margins between verdict tokens, while $\gamma(c)$ denotes the *reasoning consistency*, measured from contradiction scores between rationales generated under opposing assumptions. Decision thresholds $\alpha, \beta \in (0, 1)$ partition this two-dimensional space into four quadrants, each corresponding to a distinct verification policy. The thresholds are chosen empirically based on the distributions of $\tau(c)$ and $\gamma(c)$.

**High Certainty, High Consistency** $(\tau(c) \geq \alpha, \gamma(c) \geq \beta)$**: Direct Answering.** When the model is both decisive and logically stable, we regard its verdict as factually reliable. In this case, the system outputs a final decision directly from parametric knowledge, bypassing retrieval. This strategy prevents unnecessary external queries and yields higher efficiency without compromising accuracy.

**Low Certainty, Low Consistency** $(\tau(c) < \alpha, \gamma(c) < \beta)$**: Deep Search.** At the opposite extreme, simultaneous indecision and unstable rationales mark the claim as high risk. Such cases often involve out-of-distribution content or adversarial phrasing that disrupts coherent reasoning. We therefore escalate to a *deep search* stage in which an LLM agent iteratively generates targeted queries, retrieves and consolidates evidence, reflects to revise hypotheses, and performs reasoning before issuing a final verdict (Zhang et al., 2025).

**High Certainty, Low Consistency** $(\tau(c) \geq \alpha, \gamma(c) < \beta)$**: Targeted Search via Reasoning Consistency Signals.** In this scenario, the model assigns high probability to a verdict token but fails to provide stable reasoning under adversarial framing. Such cases often reflect *hallucinations*, where the model produces overconfident yet unreliable outputs. To mitigate this risk, we leverage reasoning consistency signals from the NLI module to guide targeted retrieval. Specifically, we identify the most contradictory rationale pairs $\{r_i^+, r_j^-\}$ and prompt the LLM to generate focused search queries aimed at retrieving evidence that addresses the hallucinated or disputed knowledge. This targeted search ensures that overconfident but logically unstable predictions are verified against external sources before issuing a final verdict.

Table 1: Overall performance of PCC and four baselines across three fact-checking benchmarks, measured by $F_1$ score. Results are reported separately for *True* and *False* labels using GPT-4o and GPT-4o-mini. Best scores in each column are highlighted in **green**.

| Framework | LLM | SciFact | | FeLMWk | | HoVER | |
|---|---|---|---|---|---|---|---|
| | | *Label = True* | *Label = False* | *Label = True* | *Label = False* | *Label = True* | *Label = False* |
| Factool | GPT-4o | 0.68 | 0.55 | 0.60 | 0.64 | 0.44 | 0.56 |
| | GPT-4o-mini | 0.65 | 0.51 | 0.48 | **0.62** | 0.41 | 0.51 |
| Factcheck-GPT | GPT-4o | 0.62 | 0.52 | 0.67 | 0.61 | 0.42 | 0.58 |
| | GPT-4o-mini | 0.58 | 0.46 | 0.55 | 0.56 | 0.22 | 0.52 |
| SAFE | GPT-4o | 0.64 | 0.53 | 0.75 | 0.65 | 0.44 | 0.68 |
| | GPT-4omini | 0.62 | 0.55 | 0.68 | 0.51 | 0.28 | 0.68 |
| FIRE | GPT-4o | 0.69 | 0.58 | 0.77 | 0.63 | 0.39 | 0.66 |
| | GPT-4o-mini | 0.67 | **0.57** | 0.71 | 0.53 | 0.30 | 0.70 |
| PCC (Ours) | GPT-4o | **0.72** | **0.62** | **0.79** | **0.68** | **0.52** | **0.76** |
| | GPT-4o-mini | **0.68** | 0.55 | **0.76** | 0.60 | **0.45** | **0.71** |

**Low Certainty, High Consistency** $(\tau(c) < \alpha, \gamma(c) \geq \beta)$**: Targeted Search via Self-Reflection.** In this quadrant, the model produces coherent rationales under both true and false assumptions but remains uncertain about the final verdict. This pattern reflects a knowledge gap: the model's reasoning is stable yet lacks the factual content needed to decisively support one side. As a result, the rationales tend to lean toward neutral relationships, offering incomplete justification for either verdict. To address this, we prompt the LLM to perform *self-reflection* on its uncertainty and generate a targeted search query that explicitly captures the missing information, enabling efficient retrieval of high-precision evidence.

## 4 EMPIRICAL FINDINGS

We evaluate the effectiveness of PCC-guided fact-checking across three challenging benchmarks. Specifically, we aim to answer the following research questions: **RQ1:** How effective is PCC-guided fact-checking compared to established baselines? **RQ2:** How well does PCC generalize across different LLM families? **RQ3:** Does PCC offer a more reliable signal than verbal confidence? **RQ4:** How sensitive is PCC to variations in model capability? **RQ5:** Do certainty and consistency signals from PCC improve retrieval and search performance?

### 4.1 EXPERIMENT SETUP

**Benchmarks and Evaluation Metric.** We evaluate PCC on three standard fact-checking datasets: SCIFACT (Wadden et al., 2020), FELMWK (Zhao et al., 2023), and HOVER (Jiang et al., 2020). Performance is reported using *macro*-$F_1$ as the primary metric, along with per-label $F_1$ scores for `True` and `False` claims. Additional dataset details are provided in Appendix C.1.

**Baselines and Models.** We compare PCC against four strong baselines: FACTOOL (Chern et al., 2023), a tool-augmented framework for factuality detection; FACTCHECK-GPT (Wang et al., 2024), an end-to-end LLM-based fact-checker that operates at multiple granularities; SAFE (Wei et al., 2024), which decomposes long-form outputs into atomic claims and verifies them iteratively through retrieval; and FIRE (Xie et al., 2025), an iterative retrieval-and-verification system with adaptive query generation, and our most directly comparable baseline. We evaluate these methods using both proprietary models (GPT-4o and Gemini-2.5) and an open-source model (Mistral-7B-Instruct).

### 4.2 RQ1: OVERALL PERFORMANCE OF PCC-GUIDED FACT-CHECKER

Table 1 compares PCC against four strong baselines on SCIFACT, FELMWK, and HOVER, evaluated using both GPT-4o and GPT-4o-mini. Across all datasets and model scales, PCC consistently outperforms the baselines under most scenarios, demonstrating its effectiveness in fact-checking.

With GPT-4o, PCC achieves the strongest performance across the board. On SCIFACT, it improves the score for the label `True` from 0.69 under FIRE to 0.72, and for the label `False` from 0.58 to

Table 2: Performance comparison on different LLM families and benchmarks.

| Framework | LLM | SciFact | | FeLMWk | | HoVER | |
|---|---|---|---|---|---|---|---|
| | | *Label = True* | *Label = False* | *Label = True* | *Label = False* | *Label = True* | *Label = False* |
| Gemini-2.5-Pro | FIRE | 0.70 | 0.54 | 0.74 | 0.71 | 0.50 | 0.72 |
| | PCC (Ours) | **0.79** | **0.65** | **0.77** | **0.74** | **0.53** | **0.75** |
| Gemini-2.5-Flash | FIRE | 0.72 | 0.58 | 0.73 | 0.70 | 0.26 | 0.73 |
| | PCC (Ours) | **0.75** | **0.60** | **0.76** | **0.72** | **0.43** | 0.70 |
| Mistral-7B-Instruct | FIRE | 0.79 | 0.57 | 0.56 | 0.52 | 0.48 | 0.58 |
| | PCC (Ours) | **0.82** | 0.53 | **0.60** | **0.55** | **0.52** | **0.61** |

0.62. On FELMWK, which requires reasoning over both encyclopedic and real-world claims, PCC reaches 0.79 for `True` and 0.68 for `False`, outperforming SAFE and FIRE by as much as five points. The most substantial relative improvement appears on HOVER, a benchmark that involves multi-hop and compositional reasoning. PCC increases the score for `False` from 0.66 under FIRE to 0.76, a relative gain of 15.2 percent, and also achieves the highest score for `True`, reaching 0.52. These results highlight PCC's strength in reducing overconfidence, a known challenge for prior methods due to the tendency of language models to overcommit to unsupported claims.

Similar trends are observed with GPT-4o-mini, though the overall scores are lower given the smaller model capacity. Nevertheless, PCC continues to outperform the baselines in most cases. On FELMWK, it achieves a score of 0.76 for `True`, compared to 0.71 under FIRE. On HOVER, it reaches 0.71 for `False`, slightly above FIRE's 0.70. These consistent gains across both high-capacity and low-capacity models underscore that PCC's advantage does not depend on model scale. Rather, its strength lies in its ability to combine internal certainty and reasoning consistency as complementary signals for reliable and adaptive fact-checking.

## 4.3 RQ2: GENERALIZATION TO OTHER LLMS

To assess whether PCC generalizes effectively across different language model families, we evaluate it on three additional LLMs: Gemini-2.5-Pro, Gemini-2.5-Flash, and Mistral-7B-Instruct. Table 2 presents the performance of PCC compared to the strongest baseline, FIRE, across SCIFACT, FELMWK, and HOVER.

Across all models and datasets, PCC consistently outperforms FIRE. For the Gemini models, PCC yields consistent gains of two to four points in $F_1$ score. On Gemini-2.5-Pro, the score for `False` claims on SCIFACT increases from 0.54 to 0.65, and on FELMWK from 0.71 to 0.74. On HOVER, PCC improves the `False` score from 0.72 to 0.75, and the `True` score from 0.50 to 0.53. Gemini-2.5-Flash shows a similar trend. Most notably, the `True` score on HOVER rises from 0.26 with FIRE to 0.43 with PCC, reflecting a substantial improvement in compositional reasoning.

The open-weight Mistral-7B model, which generally exhibits weaker calibration from verbalized confidence, shows even greater benefits from PCC. On FELMWK, the score for `False` claims increases from 0.52 to 0.55, and on HOVER, from 0.58 to 0.61. PCC also raises the `True` score on SCIFACT from 0.79 to 0.82. While the `False` score on SCIFACT decreases slightly from 0.57 to 0.53, the overall pattern across datasets remains consistently positive.

These results highlight two key takeaways. First, PCC generalizes effectively across both proprietary and open-weight language models, underscoring its model-agnostic design. Second, the most significant and consistent gains occur on claims labeled as `False`, where internal confidence signals alone tend to be less reliable. By integrating signals of certainty and consistency, PCC provides a more reliable estimate of factual confidence, enabling adaptive selection of verification strategies.

## 4.4 RQ3: HOW DOES PCC COMPARE TO VERBAL CONFIDENCE FOR FACT-CHECKING?

We compare PCC-guided verification with a baseline that uses the LLM's self-reported verbal confidence to decide when to retrieve evidence. While verbal confidence can correlate weakly with factual accuracy, it is often poorly calibrated and susceptible to overconfidence, resulting in unnecessary retrievals or missed verification opportunities.

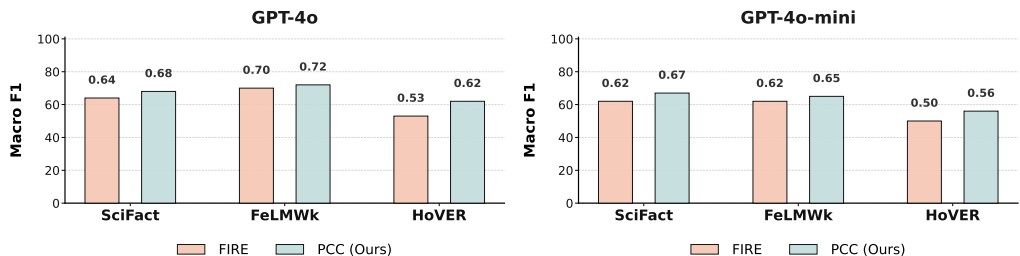

Figure 4: Macro-F$_1$ comparison of PCC versus verbal confidence on FACTOOLQA, FELMWK, and SCIFACT. PCC uses the same search module as FIRE, differing only in the confidence signal used to trigger retrieval.

As shown in Figure 4, PCC consistently outperforms FIRE across all datasets using both GPT-4o and GPT-4o-mini. These improvements are robust across model sizes and benchmark settings, confirming that combining *certainty* with *consistency* yields a more dependable retrieval signal than verbal confidence alone. The gains are especially notable on FELMWK and FACTOOLQA under GPT-4o-mini, where model confidence is generally less reliable. This improved calibration leads directly to higher macro-F$_1$ scores and more efficient retrieval usage.

## 4.5 RQ4: SENSITIVITY TO LLMs WITH DIFFERENT CAPABILITIES

To assess how much PCC benefits models of varying capacities, we compare the PCC-guided fact-checker with FIRE on the FELMWK dataset using two LLMs: GPT-4.1, a state-of-the-art model, and GPT-3.5-turbo, a significantly smaller and less capable model.

As shown in Table 3, PCC consistently outper-forms FIRE across both model scales. The improvement is particularly striking for GPT-3.5-turbo, where PCC raises the `False` score from 0.23 to 0.66. This performance is nearly on par with GPT-4.1, which achieves 0.68 with PCC. In contrast, FIRE exhibits a much larger gap between the two models, improving by 42 points for `False` claims when moving from GPT-3.5-turbo to GPT-4.1 (0.23 to 0.65). These results suggest that PCC's use of certainty and consis-

Table 3: Comparison of PCC and FIRE on FELMWK using more and less capable LLMs.

| Framework | LLM | FeLMWk | |
|---|---|---|---|
| | | *Label = True* | *Label = False* |
| FIRE | GPT-3.5-turbo | 0.70 | 0.23 |
| Ours | GPT-3.5-turbo | 0.72 | 0.66 |
| FIRE | GPT-4.1 | 0.76 | 0.65 |
| Ours | GPT-4.1 | 0.76 | 0.68 |

tency signals helps compensate for limitations in model capability. By dynamically triggering retrieval when confidence is low or reasoning is inconsistent, PCC enables more robust fact-checking that is less dependent on the raw power of the underlying LLM.

## 4.6 RQ5: DO CERTAINTY AND CONSISTENCY SIGNALS FROM PCC IMPROVE SEARCH?

In the targeted search setting, queries are constructed from contradictory rationale pairs, focusing retrieval on parts of the claim where the model shows the greatest uncertainty.

To evaluate the effectiveness of targeted retrieval guided by PCC, we compare it against a deep search base-line that performs generic multi-hop retrieval without leveraging contradiction signals. As shown in Table 4, results across SciFact, FeLMWk, and HoVER reveal a clear quadrant-specific pattern.

Table 4: Comparison of targeted retrieval versus deep search across different confidence–consistency regions.

| Region | SciFact | | FeLMWk | | HoVER | |
|---|---|---|---|---|---|---|
| | *Deep* | *Targeted* | *Deep* | *Targeted* | *Deep* | *Targeted* |
| Hi Cert. & Low Cons. | 0.58 | **0.66** | 0.55 | **0.67** | 0.55 | **0.57** |
| Low Cert. & Hi Cons. | 0.61 | **0.65** | **0.67** | 0.62 | 0.46 | **0.52** |

In the high certainty, low consistency region, targeted search consistently outperforms deep search, achieving gains of up to twelve points in macro-F$_1$. For instance, performance on SciFact improves from 0.58 to 0.66, and on FeLMWk from 0.55 to 0.67. These results suggest that when the model is confident yet offers unstable justifications, contradiction-driven query generation helps surface missing evidence and mitigates confidently incorrect predictions.

In contrast, deep search tends to be more effective in the low certainty, high consistency region. On FeLMWk, deep search reaches 0.67 compared to 0.62 with targeted search, and on SciFact, 0.65 compared to 0.61. In these cases, although the model exhibits weaker confidence, its reasoning remains coherent, and broad retrieval proves more useful than narrowly focused contradiction-based queries. On HoVER, which requires more compositional and multi-hop reasoning, the same trend holds, albeit with lower absolute scores. Targeted search improves performance in the high-certainty, low-consistency region (0.57 versus 0.55), while deep search is more effective in the low-certainty, high-consistency region (0.52 versus 0.46).

These findings indicate that PCC's dual signals guide not only whether to retrieve but also how to retrieve. Reasoning consistency highlights when contradiction-driven search is likely to help, while low-certainty signals favor wider evidence exploration. By enabling more context-aware retrieval strategies than fixed-depth baselines, PCC supports the development of adaptive, self-reflective fact-checking agents.

## 5 RELATED WORK

**Fact-Checking with LLMs** Fact-checking aims to determine whether a claim is factually correct, typically by verifying it against retrieved evidence (Thorne & Vlachos, 2018). While traditional supervised approaches have made progress, recent work increasingly adopts LLM-based methods (Manakul et al., 2023; Pan et al., 2023; Wang & Shu, 2023; Fadeeva et al., 2024; Tang et al., 2024; Wang et al., 2025). For example, FactCheck-GPT (Wang et al., 2024), Self-Checker (Li et al., 2024), and FActScore (Min et al., 2023) use LLMs directly for verification, bypassing conventional pipeline stages. SAFE (Wei et al., 2024) addresses long-form factuality by decomposing responses into atomic claims and verifying each through multi-step reasoning, including search and evidence assessment. FIRE (Xie et al., 2025) reduces retrieval cost by adaptively choosing between answering and searching, guided by the model's self-reported (verbal) confidence. However, verbal confidence is often poorly calibrated and prone to overconfidence. We propose a more reliable alternative by jointly modeling *internal certainty* and *reasoning consistency*. These complementary signals enable more reliable confidence estimation and guide an adaptive verification strategy.

**Uncertainty Quantification in LLMs** Factual confidence (Liu et al., 2025; Geng et al., 2024) denotes an LLM's estimated likelihood that its output is correct. Prior methods include self-reported *verbalization*, though often overconfident (Xiong et al., 2023; Zhao et al., 2024b); likelihood-based measures such as *sequence probability* and *surrogate tokens*; and auxiliary *probes* on hidden states (Mahaut et al., 2024) or pre-trained heads (Shelmanov et al., 2025; Vazhentsev et al., 2024). Other work explores response diversity (Portillo Wightman et al., 2023), semantic entropy under distribution shift (Kuhn et al., 2023), reasoning topology (Da et al., 2025), entailment graph (Da et al., 2024), reflection-based prompting (Zhao et al., 2024a), self-certainty (Kang et al., 2025), and calibration via structured formats or alignment with token probabilities (Kadavath et al., 2022; Kumar et al., 2024; Detommaso et al., 2024; Vazhentsev et al., 2024). Yet recent studies show models remain prone to miscalibration and struggle to explicate their own uncertainty (Kirchhof et al., 2025). We address these challenges with *PCC*, which combines *internal certainty* from log-probability margins with *reasoning consistency* from NLI-based contradiction signals, yielding a robust and interpretable confidence measure transferable across LLMs (Farquhar et al., 2024).

## 6 CONCLUSION AND DISCUSSION

We introduced *Probabilistic Certainty and Consistency (PCC)*, a framework for estimating the factual confidence of LLMs by combining internal certainty, measured through the probabilistic margin of the predicted verdict token, with reasoning consistency, assessed via contradiction logits from natural language inference models applied to adversarially framed rationales. By leveraging these two complementary signals, the PCC-guided fact-checker adaptively selects the most appropriate verification strategy for each claim. Experiments on three challenging benchmarks demonstrate its effectiveness. Ablation studies further validate that certainty and consistency are both complementary and broadly generalizable across model families and capability levels, and that jointly modeling these signals enables more targeted and effective retrieval.

ETHICS STATEMENT

This work adheres to the principles of responsible AI research and development. Our objective is to enhance the reliability, transparency, and trustworthiness of LLMs, with a particular focus on factual confidence estimation and adaptive fact-checking. We acknowledge that LLMs may still produce biased or inaccurate outputs due to artifacts in their pretraining data or biases in external retrieval sources. Importantly, our system is intended to assist, not replace, human decision-making, especially in contexts where factual accuracy is critical. This study does not involve human subjects, sensitive personal data, or privacy-sensitive information.

REPRODUCIBILITY STATEMENT

**Code & Data Availability.** We will release the codebase upon acceptance. All datasets used in our experiments are publicly available.

**Compute Resources & Cost.** Experiments were conducted using a combination of local GPUs and cloud-based APIs. For proprietary models, we relied on their official APIs and consistently used the cost-effective versions within each model family. Evaluation with open-source models required minimal compute, remaining feasible on a single modern GPU.

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

# Appendix

## Table of Contents

## A  DISCLOSURE OF LLM USAGE

LLMs were used solely to aid or polish writing. Specifically, we employed LLMs to refine grammar, improve phrasing, and enhance the overall readability of the manuscript. No LLM contributed to research ideation, experimental design, data analysis, or substantive content creation.

## B  IMPACT, LIMITATIONS, AND FUTURE IMPROVEMENTS

Our proposed framework, *Probabilistic Certainty and Consistency (PCC)*, offers a reliable and generalizable approach to uncertainty estimation across diverse LLMs, with several key implications for fact-checking systems:

- **Controlling resource usage:** Fact-checking pipelines often over-rely on costly retrieval. By jointly modeling internal certainty and reasoning consistency, PCC allows the system to decide when retrieval is necessary and when the model's parametric knowledge is sufficient. This enables fine-grained control over computational cost, improving efficiency and scalability in real-world deployments.
- **Enhancing robustness:** Overconfident yet incorrect predictions remain a critical challenge in LLM-based fact-checking. PCC mitigates this issue by cross-validating certainty with consistency, reducing susceptibility to hallucinations, and improving the reliability of factual predictions.
- **Improving interpretability:** A major barrier to trust in LLM-based verification is the opacity of confidence estimates. PCC addresses this by decomposing confidence into *certainty* and *consistency*, exposing failure modes such as "overconfident hallucinations" and offering interpretable diagnostics into the model's factual judgment.

Despite these contributions, PCC has several limitations. First, it relies on fixed thresholds for routing decisions; while effective in our experiments, such thresholds may not generalize across all models or domains. Second, PCC requires access to token-level log probabilities, which limits its

applicability to APIs that do not expose this information. Third, the current framework is focused solely on textual fact-checking, leaving its performance in multimodal unexplored.

Future work can address these challenges in several directions. Replacing fixed thresholds with adaptive or learned policies may enable more flexible and model-agnostic routing. Incorporating richer reasoning signals, such as causal inference or factual entailment chains, could further enhance robustness. Applying PCC in scientific and medical domains, where calibrated trust is essential, may uncover domain-specific failure patterns and inform high-stakes applications. Finally, extending PCC to multimodal settings would broaden its utility for fact-checking in complex, real-world information environments.

## C DETAILED EXPERIMENTAL SETTINGS

### C.1 DATASET DETAILS

We evaluate our framework on three challenging fact-checking datasets: SciFact (Wadden et al., 2020), FeLMWk (Zhao et al., 2023), and HoVER (Jiang et al., 2020). Each dataset consists of natural language claims paired with ground-truth factual labels (`True` or `False`). Below we summarize the key characteristics and preprocessing choices for each dataset:

- **SciFact:** This dataset contains expert-written scientific claims derived from PubMed abstracts, each paired with annotated evidence. The original evidence-level labels include `Support`, `Refute`, and `Not Enough Info`. To ensure consistency, we convert this to a binary claim-level format by (i) filtering out claims with contradictory labels across evidence and (ii) excluding claims associated with `Not Enough Info`. Following prior work (Xie et al., 2025) and to control API costs, we sample 187 claims for our experiments.

- **FeLMWk:** This dataset consists of claims generated by LLMs across diverse domains, annotated with fine-grained factuality labels. For our evaluation, we select the subset of claims that require world knowledge for verification, which is the closest setting to fact-checking.

- **HoVER:** This dataset contains multi-hop claims that require reasoning over multiple pieces of evidence. Each claim is associated with a Wikipedia passage set, and the difficulty increases with hop length. We focus on the most challenging 4-hop claims, which demand reasoning over several interconnected evidence spans. To reduce API costs while preserving difficulty, we sample 190 claims for evaluation.

### C.2 IMPLEMENTATION DETAILS

Our method requires access to token-level log probabilities from LLMs. For OpenAI models, these are directly accessible via the API. Open-source models such as Mistral also provide log probabilities through their standard inference APIs. In contrast, the native Gemini API does not expose token log probabilities; to address this, we use Gemini via VertexAI[1], which supports log-probability outputs. For web search, we rely on the Serper API[2], which provides access to Google Search results. For deep retrieval, we adopt an off-the-shelf deep search agent implementation[3] built on top of LangGraph, which supports multi-hop query refinement and evidence aggregation. For reasoning consistency, we use a pre-trained NLI model [4].

## D EVALUATION OF FACTUAL CONFIDENCE ESTIMATION

We evaluate confidence estimation by comparing a model's reported probabilities with its empirical correctness. Let $\mathcal{D} = \{(x^{(i)}, y^{(i)})\}_{i=1}^{N}$ be a dataset of $N$ claims with ground-truth labels $y^{(i)} \in \mathcal{Y} = \{1, \ldots, K\}$. For each input $x^{(i)}$, the model $\phi$ outputs a predicted label $\hat{y}^{(i)} = \phi(x^{(i)})$ along with

---

[1] https://cloud.google.com/generative-ai-studio
[2] https://serper.dev/
[3] https://github.com/google-gemini/gemini-fullstack-langgraph-quickstart
[4] https://huggingface.co/MoritzLaurer/DeBERTa-v3-base-mnli-fever-anli

a confidence score $p^{(i)} \in [0, 1]$ representing the estimated probability that the prediction is correct. We define the correctness indicator

$$z^{(i)} = \mathbf{1}\Big(\hat{y}^{(i)} = y^{(i)}\Big),$$

which equals $1$ if the prediction is correct and $0$ otherwise.

## D.1 EXPECTED CALIBRATION ERROR (ECE)

A model is *well-calibrated* if its predicted probabilities match empirical correctness frequencies. To quantify miscalibration, we partition the interval $[0, 1]$ into $M$ equal-width bins. Let

$$\mathcal{I}_m = \big\{ i \mid p^{(i)} \in \big( \tfrac{m-1}{M}, \tfrac{m}{M} \big] \big\}$$

be the set of indices whose confidence values fall into bin $m$. For each bin, we compute the average predicted confidence and empirical accuracy:

$$\text{Conf}(m) = \frac{1}{|\mathcal{I}_m|} \sum_{i \in \mathcal{I}_m} p^{(i)}, \qquad \text{Acc}(m) = \frac{1}{|\mathcal{I}_m|} \sum_{i \in \mathcal{I}_m} z^{(i)}.$$

The expected calibration error (ECE) is then defined as

$$\text{ECE}(p, z) = \sum_{m=1}^{M} \frac{|\mathcal{I}_m|}{N} \big| \text{Acc}(m) - \text{Conf}(m) \big|.$$

A smaller ECE indicates closer alignment between predicted probabilities and true correctness rates. While widely used, ECE has limitations: it depends on the bin count $M$, and in high-accuracy settings a degenerate predictor that outputs uniformly high confidence can yield deceptively low values. Nevertheless, we report ECE due to its interpretability and its role as a standard baseline in prior work.

## D.2 AREA UNDER THE ROC CURVE (AUROC)

Calibration does not capture whether confidence scores *rank* predictions correctly. To measure discrimination ability, we compute the ROC curve. For a threshold $\tau \in [0, 1]$, define

$$\text{TPR}(\tau) = \frac{\sum_{i=1}^{N} \mathbf{1}(p^{(i)} \geq \tau \wedge z^{(i)} = 1)}{\sum_{i=1}^{N} \mathbf{1}(z^{(i)} = 1)}, \quad \text{FPR}(\tau) = \frac{\sum_{i=1}^{N} \mathbf{1}(p^{(i)} \geq \tau \wedge z^{(i)} = 0)}{\sum_{i=1}^{N} \mathbf{1}(z^{(i)} = 0)}.$$

Plotting $\text{TPR}(\tau)$ against $\text{FPR}(\tau)$ as $\tau$ varies yields the ROC curve. An ideal confidence estimator achieves $\text{TPR} = 1$ while keeping $\text{FPR} = 0$, producing a curve that passes through the top-left corner, whereas random confidence scores generate a diagonal line from $(0, 0)$ to $(1, 1)$.

The *area under the ROC curve* (AUROC) summarizes the ROC curve as

$$\text{AUROC}(p, z) = \int_0^1 \text{TPR}(\alpha) \, d\alpha,$$

where $\alpha$ denotes the false positive rate. AUROC ranges from $0.5$ (no discriminative power) to $1.0$ (perfect separation). Higher AUROC indicates that the confidence function assigns systematically larger scores to correct predictions than to incorrect ones. We compute AUROC empirically from the ROC curve constructed using confidence–label pairs $(p^{(i)}, z^{(i)})$.

## E ADDITIONAL EXPERIMENTAL RESULTS

### E.1 ECE RESULTS

Table 5 reports the Expected Calibration Error (ECE) across three datasets: SciFact, FeLMWk, and HoVER, using both proprietary and open-source LLMs. We compare four confidence estimation methods: (i) *Verbal* self-reported confidence, (ii) *Certainty* based on token-level log-probability

Table 5: **Expected Calibration Error (ECE)** across three datasets and six LLMs (lower is better).

| LLM | SciFact | | | | FeLMWk | | | | HoVER | | | |
|---|---|---|---|---|---|---|---|---|---|---|---|---|
| | Verbal | Cert. | Cons. | PCC | Verbal | Cert. | Cons. | PCC | Verbal | Cert. | Cons. | PCC |
| GPT-4o | 0.3712 | 0.3677 | 0.3001 | 0.3345 | 0.1961 | 0.2204 | 0.1736 | 0.1732 | 0.4000 | 0.4068 | 0.2732 | 0.3038 |
| GPT-4o-mini | 0.3634 | 0.3225 | 0.2347 | 0.2101 | 0.3441 | 0.3271 | 0.2292 | 0.2395 | 0.3937 | 0.3791 | 0.1666 | 0.2076 |
| Gemini-2.5-Pro | 0.2353 | 0.2746 | 0.1062 | 0.1291 | 0.2024 | 0.2644 | 0.2183 | 0.1781 | 0.3830 | 0.4275 | 0.2833 | 0.3017 |
| Gemini-2.5-Flash | 0.3309 | 0.1943 | 0.2932 | 0.1989 | 0.2277 | 0.1946 | 0.2200 | 0.1522 | 0.4542 | 0.2607 | 0.2538 | 0.2403 |
| Mistral-7B | 0.2660 | 0.2814 | 0.1243 | 0.1346 | 0.4552 | 0.4183 | 0.2918 | 0.3287 | 0.4691 | 0.4758 | 0.2912 | 0.3156 |

margins, (iii) *Consistency* derived from cross-assumption NLI judgments, and (iv) the proposed *PCC* combination.

Overall, PCC consistently achieves the lowest ECE across nearly all settings, demonstrating that it provides better-calibrated confidence estimates than either verbal self-reports or single-signal baselines. The improvements are most pronounced on challenging datasets such as HoVER, where retrieval is crucial, and on open-weight models such as Mistral-7B, which typically exhibit poor calibration. These results suggest that jointly modeling certainty and consistency yields more reliable factual confidence estimates, improving robustness across both model families and task domains.

### E.2 SCORE DISTRIBUTION

We further analyze the distribution of confidence scores produced by different methods—verbal confidence, internal certainty, reasoning consistency, and PCC—on SciFact, FeLMWk, and HoVER. As shown in Figure 5, PCC produces the sharpest separation between correct and incorrect predictions. For correct cases, PCC scores are concentrated in the high-confidence region ($[0.7, 1.0]$), while incorrect cases shift toward lower values, yielding a clear margin around the decision threshold. In contrast, verbal confidence exhibits substantial overlap between correct and incorrect predictions, reflecting the tendency of LLMs to report overconfident but unreliable scores. Internal certainty and reasoning consistency individually offer partial separation, but PCC's joint modeling yields the most discriminative distributions.

### E.3 ROC ANALYSIS

To evaluate discriminative ability, we treat each confidence signal as a binary classifier over correctness by sweeping a threshold across the score range and computing the receiver operating characteristic (ROC) curve along with the corresponding area under the curve (AUC). Across SciFact, FeLMWk, and HoVER, PCC achieves the highest AUC values, indicating that it more effectively ranks correct predictions above incorrect ones. The improvement is especially pronounced on HoVER—the most compositional benchmark—where hallucinations and spurious reasoning shortcuts are prevalent. In this setting, reasoning consistency provides additional signal, enabling PCC to separate correct from incorrect predictions more reliably than verbal confidence.

### E.4 ECE ANALYSIS

We compute Expected Calibration Error (ECE) for each confidence signal using equal-width binning with $K$ bins (default $K=15$). For *verbal confidence*, we directly take the model-reported scalar. For *internal certainty*, we transform the two-token log-probability margin into a probability using the logistic mapping described earlier. For *reasoning consistency*, we use the NLI-derived score $\gamma(c)$, and for *PCC*, we evaluate the combined score $\mathrm{H}(m, \gamma)$.

Across all datasets and models, PCC consistently achieves the lowest ECE, with especially pronounced improvements on `False` claims, where overconfident errors are most common. Reliability diagrams (Figure 7) confirm these findings: the PCC curves lie closest to the diagonal identity line, indicating that predicted probabilities better reflect empirical correctness frequencies.

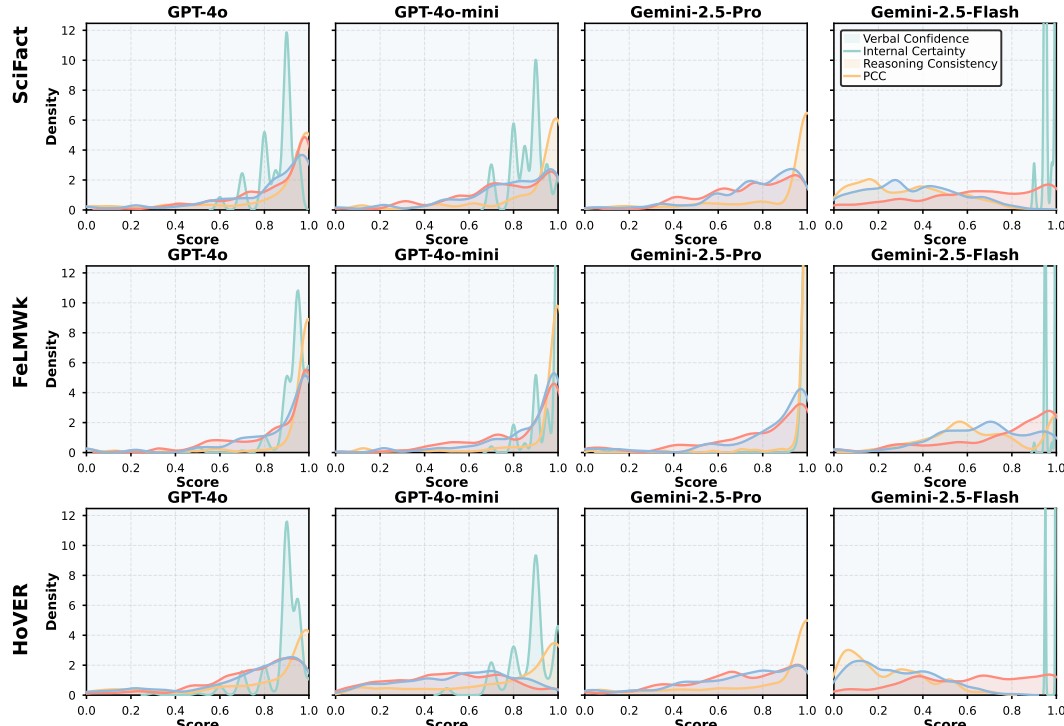

Figure 5: Kernel density estimation (KDE) plots of score distributions for correct (green) and incorrect (red) predictions across datasets. PCC yields the clearest separation, reducing overlap in the overconfident region.

# F  PROMPT TEMPLATES

In this section, we present the full set of prompt templates used in our framework. Specifically, Prompt 1 (Figure 8) is used for verbal confidence elicitation, Prompt 2 (Figure 9) for internal certainty estimation, Prompt 3 (Figure 10) for reasoning consistency estimation, Prompt 4 (Figure 11) for targeted search leveraging contradiction signals from reasoning consistency, Prompt 5 (Figure 12) for reflection-based query generation, and Prompt 6 (Figure 13) for deep search. The exact templates are provided in the figures below.

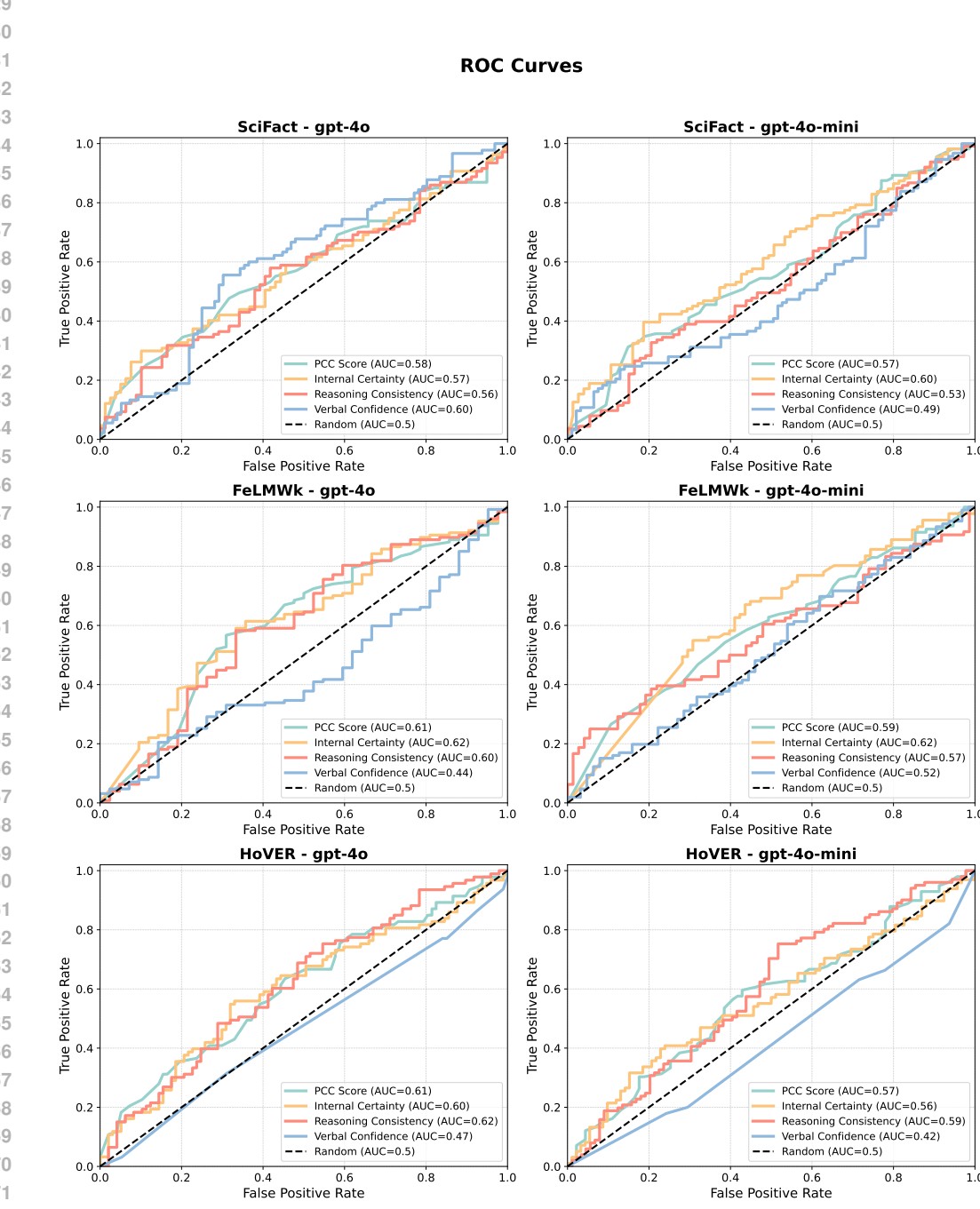

Figure 6: ROC curves comparing verbal confidence, internal certainty, reasoning consistency, and PCC. PCC consistently dominates the other methods, achieving the largest separation on HoVER where compositional reasoning is required.

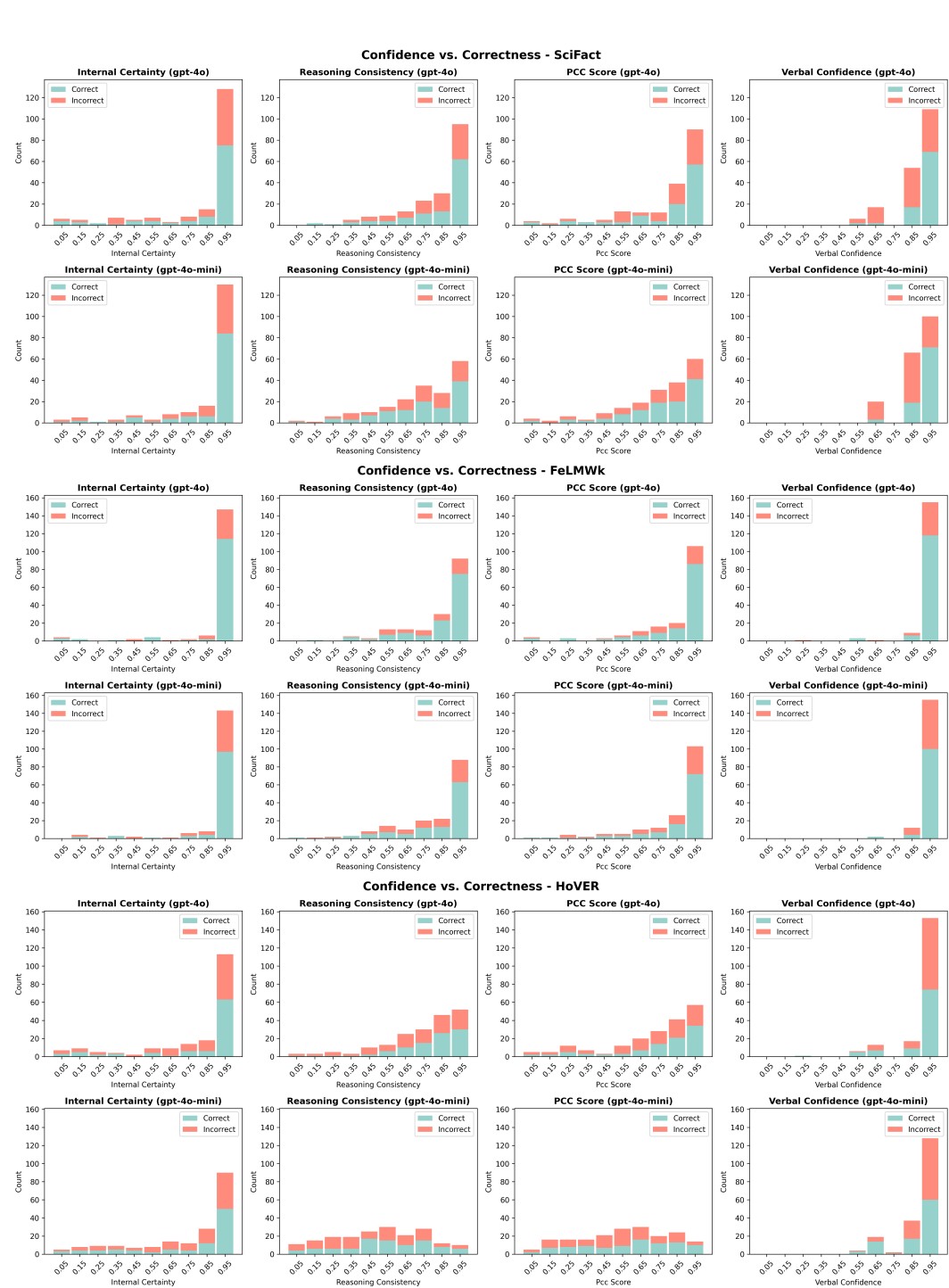

Figure 7: Reliability diagrams for verbal confidence, internal certainty, reasoning consistency, and PCC across datasets. PCC exhibits curves closest to the diagonal, demonstrating improved calibration and reduced overconfidence.

**Prompt Template 1: Verbal Confidence for ECE**

```
Instructions:
1. You are provided with a STATEMENT.
2. Your goal is to assess the factual accuracy of the STATEMENT.
3. Before presenting your final answer, think step-by-step and show
    your reasoning.
4. Your final answer should be either "{_FACTUAL_LABEL}" or "{
    _NON_FACTUAL_LABEL}".
5. After your reasoning, provide your confidence in your answer as a
      number between 0 and 1,
    where 1 means absolutely certain and 0 means completely uncertain
       .
6. Format your final answer as a JSON object in the following
     structure:
{
  "final_answer": "{_FACTUAL_LABEL}" or "{_NON_FACTUAL_LABEL}",
  "confidence": <float between 0 and 1>
}

STATEMENT:
{_STATEMENT_PLACEHOLDER}
```

Figure 8: Prompt template for verbal confidence ECE.

**Prompt Template 2: Internal Certainty**

```
Instructions:
1. You are provided with a STATEMENT.
2. Your goal is to assess the factual accuracy of the STATEMENT.
3. Before presenting your final answer, think step-by-step.
4. Your final answer should be either \"True\" or \"False\".
5. Return only the name of the label, and nothing else.

STATEMENT: {_STATEMENT_PLACEHOLDER}
```

Figure 9: Prompt template for internal certainty.

**Prompt Template 3: Reasoning Consistency**

```
{
    "true": (
        "You are a knowledgeable expert. Strongly support the
            following claim with a concise, factual argument. "
        "Focus only on supporting evidence. Avoid hedging.\nClaim: {
            claim}"
    ),
    "false": (
        "You are a critical skeptic. Strongly refute the following
            claim with a concise, factual argument. "
        "Focus only on refuting evidence. Avoid hedging.\nClaim: {
            claim}"
    ),
}
```

Figure 10: Prompt template for reasoning consistency.

**Prompt Template 4: Targeted Search Prompt**

```
The following two statements strongly contradict each other
    regarding the factual accuracy of a claim:

Premise: \"{_PREMISE_PLACEHOLDER}\"

Hypothesis: \"{_HYPOTHESIS_PLACEHOLDER}\"

Your task: Based on the specific disagreement between these two
    rationales, write a concise, targeted web search query that
    would help resolve the factual conflict.
Return only the search query.
```

Figure 11: Prompt template for targeted search.

**Prompt Template 5: Reflection Prompt**

```
Instructions:
1. You are given a STATEMENT.
2. After analysis, you found that both \"True\" and \"False\"
    explanations for the STATEMENT are logically consistent, but
    your confidence is low.
3. Your task is to suggest a focused search query or keywords that
    could help find information to resolve the uncertainty.
4. Format your answer as a JSON object with the following field:
{{
    \"search_query\": \"<suggested search query or keywords>\"
}}

STATEMENT:
{_STATEMENT_PLACEHOLDER}
```

Figure 12: Prompt template for reflection.

---

**Prompt Template 6: Deep Search**

```
Instructions:
1. You are provided with a STATEMENT.
2. Your goal is to assess the factual accuracy of the STATEMENT.
3. Before presenting your final answer, think step-by-step and show
   your reasoning.
4. Your final answer should be either \"{_FACTUAL_LABEL}\" or \"{
   _NON_FACTUAL_LABEL}\".
5. Format your final answer as a JSON object in the following
   structure:
{{
    \"final_answer\": \"{_FACTUAL_LABEL}\" or \"{_NON_FACTUAL_LABEL
       }\"
}}

STATEMENT:
{_STATEMENT_PLACEHOLDER}
```

Figure 13: Prompt template for deep search.

