# OpenReview forum: "Fact-Checking with Large Language Models via Probabilistic Certainty and Consistency"
_ICLR.cc/2026/Conference — ICLR 2026 Conference Withdrawn Submission_

### Official Review · Reviewer_2cNN · 2025-10-15

**Soundness:** 1
**Presentation:** 2
**Contribution:** 2
**Rating:** 2
**Confidence:** 3

**Summary:**

The paper works towards LLM-based fact-checking, aiming to improve performance and make it more efficient. This is based on an uncertainty-guided mechanism to decide how to fact-check a statement: Either through a deep search, a targeted search, or by directly answering from the fact-checking LLM’s parametric knowledge.

**Strengths:**

1. The method is tested on a good amount of datasets (3), LLMs (5, including frontier LLMs), and against 3 baselines.
2. The paper promises to release code upon acceptance for reproducibility.

**Weaknesses:**

Weaknesses, in order of magnitude:
1. Some statements in the paper are false:
    1. "Across all datasets and models, PCC consistently yields the lowest ECE (Figure 3).", but in Figure 3, PCC is only the best in 4/15 cases.
    2. “harmonically combining them [internal certainty and reasoning consistency] yields consistently better calibration.”, actually reasoning consistency has strictly better ECE than the harmonic combination (PCC?) in 8/15 cases
    3. “Across Sci-Fact, FeLMWk, and HoVER, PCC achieves the highest AUC values”, again this is not true according to Figure 6.
2. Other claims are illogical
    1. “verbal confidence is systematically overconfident and poorly aligned with accuracy”, but only the ECE was measured, which can be high both due to under and overconfidence.

    2. “internal certainty and reasoning consistency each capture only partial signals”, how can this be followed from the previous ECE plots?
    3. There is often the logical fallacy of “The overall performance is better, hence the decision strategy of when to apply which strategy must be better” (e.g., lines 413-415). As explained above and below, the results indicate the decision strategy is very noisy. The more likely hypothesis in my eyes is that the overall performance is better simply because we mix-in more inference-time compute (deep search) in some cases. In any case, there are two potential hypotheses for why the overall performance is better (if it is), and to conclude with one of them requires to run ablation experiments.
3. The results are quite weak:
    1. The uncertainty to do the fact checking has an AUROC of at best 0.62 (0.5 is random) in Figure 6. Hence, it is likely that the method just randomly decides whether to rely on internal knowledge, do a “deep search” or a “targeted search”
    2. The overall performance (i.e., deciding which strategy to apply and then applying it) is often times only marginally better than baselines, and no error bars are reported. This is although error bars would be important especially for Fact Checking, since the benchmarks for this are relatively noisy.
4. The paper in its current state is very unclear:
    1. In Section 2.3, PCC is benchmarked, and argued to be one of the best methods. But PCC was never introduced in Section 2.2 or 2.1. Is PCC Internal Certainty multiplied with Reasoning consistency?
    2. I cannot understand Figure 5 at all, the legend (and caption) seem to be different from what is actually being shown.
    3. In Section 3, how is the deep search implemented, and how is self-reflection implemented? Is this just using the prompts in the appendix to prompt GPT 4o differently (including web search)?

### Weaknesses that did not influence my score and don’t need rebuttal but would be great to fix in the revised version

1. “derived from log-probability margins between verdict tokens”, actually it is derived from probabilities, not log probs, according to the first equation
2. Please number your equations

**Questions:**

In Section 2.1, tau(c) is a half deterministic: If the top two tokens are equal, then you give certainty 1, otherwise you base it off the probabilities of True and False tokens in the vocabulary. What happens if you remove the first bit and just always use the probabilities?

---

> ### Author Response · Authors · 2025-11-21
>
> We thank the reviewer for the thoughtful comment and constructive feedback. We greatly appreciate the time and effort you have invested in reviewing our work. Below, we address each of your concerns in detail.
>
>
> **Q1. Some statements are false.**
>
> Thank you for your attention to detail. We agree that these statements are confusing, and we will reframe them as follows:
> - Compared to verbal confidence, PCC consistently yields the lowest ECE across all datasets and models.
> - Harmonically combining internal certainty and reasoning consistency yields better calibration compared to verbal confidence.
> - Across Sci-Fact, FeLMWk, and HoVER, PCC achieves better AUC values than verbalized confidence, except on Sci-Fact using GPT-4o.
>
>
> **Q2. Other claims are illogical.**
>
> Thank you for pointing out. We agree that some statements are not accurate and need to be revised.
> - This statement is from observation of previous works, rather than derived from our experiment results. We will reframe this and add the following works [1, 2].
> - As shown in Figure 3, internal certainty and reasoning consistency alone do not perform uniformly the best across all scenarios compared to verbal confidence. However, PCC combines both signals and can outperform verbal confidence in all scenarios.
> - We agree there could be multiple hypotheses associated with the performance gain, and we try our best to provide ablation studies to separate them. As other reviewers have pointed out, we are working on adding a comparison that uses only internal certainty and reasoning consistency.
>
> [1] Matéo Mahaut, Laura Aina, Paula Czarnowska, Momchil Hardalov, Thomas Müller, and Lluis Marquez. 2024. Factual Confidence of LLMs: on Reliability and Robustness of Current Estimators. In Proceedings of the 62nd Annual Meeting of the Association for Computational Linguistics (Volume 1: Long Papers), pages 4554–4570, Bangkok, Thailand. Association for Computational Linguistics.
>
> [2] Miao Xiong, Zhiyuan Hu, Xinyang Lu, YIFEI LI, Jie Fu, Junxian He, & Bryan Hooi (2024). Can LLMs Express Their Uncertainty? An Empirical Evaluation of Confidence Elicitation in LLMs. In The Twelfth International Conference on Learning Representations.
>
>
> **Q3. The results are quite weak.**
>
> The results should be viewed comparatively against the baseline (verbal confidence). Additionally, the results are on par with other works on evaluating confidence using AUC, such as Figure. 3 in (Xiong et al., 2024) [2].
>
> Thank you for the suggestion on adding error bars. Could you elaborate on “This is although error bars would be important especially for Fact Checking, since the benchmarks for this are relatively noisy.”?
>
>
> **Q4. The paper is very unclear.**
>
> Thank you for bringing this to our attention. We aim to clarify them below:
> - As described in the introduction paragraph of Section 2 and Figure 2, PCC uses both internal certainty and reasoning consistency signals, and profiles each claim into four distinct regions.
> - Figure 5 shows the score distribution of Kernel Density Estimation (KDE), we made a typo in the legend. The four different colors reflect each type of confidence estimation.
> - For deep search, we use an off-the-shelf deep search agent, as described in Section C.2. For self-reflection, we have listed the prompt in Figure 13.

---

### Official Review · Reviewer_fhBk · 2025-10-31

**Soundness:** 2
**Presentation:** 3
**Contribution:** 2
**Rating:** 4
**Confidence:** 4

**Summary:**

This paper addresses the unreliability of LLMs' verbal confidence in fact-checking by proposing Probabilistic Certainty and Consistency (PCC), which combines internal certainty from token probabilities with reasoning consistency from NLI-scored contradictions between adversarially-prompted rationales. Experiments across three benchmarks with multiple LLMs show PCC provides reliable confidence estimates.

**Strengths:**

- The motivation is clear and addresses a real problem. The paper identifies that verbal confidence is poorly calibrated across models and provides a reasonable human analogy for why combining decisiveness with reasoning stability might help.
- The proposed two-dimensional confidence framework offers conceptual value. Decomposing confidence into internal certainty and reasoning consistency is interpretable and could potentially reveal different types of model uncertainty.
- The paper is well-organized and easy to follow. The presentation is systematic, and the figures effectively illustrate the core ideas.

**Weaknesses:**

I do have some concerns with the paper that I believe should be addressed:

- The authors mention that thresholds α and β are chosen empirically based on the distributions, but they provide no concrete procedure for selecting these values, no sensitivity analysis showing how performance varies with different thresholds, and no solution to the acknowledged generalization problem, which makes me question how we would actually deploy this method in new domains.
- The paper uses only 187 samples from SciFact and 190 from HoVER, which raises concerns about the statistical significance of the reported improvements and the generalizability of the findings. I wonder why the authors didn't validate their approach on larger-scale QA benchmarks, such as TruthfulQA or CommonsenseQA, which would provide more robust evidence.
- The paper emphasizes in the Abstract that PCC improves efficiency. But I also noticed that the method requires generating 2*K rationales and performing K² NLI inferences per claim. At the same time, the authors report no actual runtime measurements or cost comparisons, making the efficiency claims hard to believe.
- The entire reasoning consistency framework hinges on DeBERTa-v3-base-mnli-fever-anli accurately detecting contradictions; however, this paper provides no validation that this NLI model performs reliably in fact-checking contexts, meaning that any errors in the NLI component directly propagate into confidence estimates.
- Key ablations are missing. I don't see a comparison with a baseline that uses token-level probabilities without the adversarial prompting framework, which makes it impossible to isolate whether the gains come from moving away from verbal confidence or specifically from the consistency modeling.

**Questions:**

Please see the weaknesses I've outlined.

---

> ### Author Response · Authors · 2025-11-21
>
> We thank the reviewer for the thoughtful comment and constructive feedback. We greatly appreciate the time and effort you have invested in reviewing our work. Below, we address each of your concerns in detail.
>
>
> **Q1. Lack of sensitivity analysis on thresholds.**
>
> Thank you for raising this concern. We agree that including a sensitivity analysis over different threshold cutoffs is important for understanding the robustness of the routing decisions. We are working on adding this analysis in the next version of our paper.
>
> **Q2. Concerns about the dataset size not being significant.**
>
> To control the cost of API calls, prior work in LLM-based fact-checking has also used datasets of comparable scale. For example, FIRE [1] and FactCheck-Bench [2] evaluate on similarly sized subsets due to the computational demands of multi-hop retrieval and LLM-based verification. We therefore follow this established practice.
>
>
> [1] Zhuohan Xie, Rui Xing, Yuxia Wang, Jiahui Geng, Hasan Iqbal, Dhruv Sahnan, Iryna Gurevych, and Preslav Nakov. 2025. FIRE: Fact-checking with Iterative Retrieval and Verification. In Findings of the Association for Computational Linguistics: NAACL 2025, pages 2901–2914, Albuquerque, New Mexico. Association for Computational Linguistics.
>
> [2] Yuxia Wang, Revanth Gangi Reddy, Zain Muhammad Mujahid, Arnav Arora, Aleksandr Rubashevskii, Jiahui Geng, Osama Mohammed Afzal, Liangming Pan, Nadav Borenstein, Aditya Pillai, Isabelle Augenstein, Iryna Gurevych, and Preslav Nakov. 2024. Factcheck-Bench: Fine-Grained Evaluation Benchmark for Automatic Fact-checkers. In Findings of the Association for Computational Linguistics: EMNLP 2024, pages 14199–14230, Miami, Florida, USA. Association for Computational Linguistics.
>
>
> **Q3. Concerns about efficiency.**
>
> Thank you for raising this concern. Our method introduces additional API calls only for claims that fall outside the “high certainty and high consistency” region. For claims within that region, the model directly verifies the claim without invoking web search or additional LLM calls, which reduces overall cost. We are currently preparing a cost analysis comparing PCC routing to baseline pipelines and will include these measurements in the next version.
>
>
> **Q4. Reasoning consistency performance depends on the pre-trained NLI model**
>
> Thank you for raising this concern. We use the NLI model as a proxy for reasoning consistency, rather than entirely relying on its classification results. We take the logit of the contradiction label as a signal for how strongly the assume-true and assume-false claims refute each other, rather than take the label of support, refute, or neutral as the signal. Therefore, even though the NLI model's performance may cause error propagation, its effect should be less than taking the classification result directly from the NLI model.
>
>
> **Q5. Missing comparison against baseline that only uses token-level probabilities as the signal for routing.**
>
> Thank you for bringing up this point. We are working on adding a comparison that uses token-level probability as the sole signal for selecting verification strategies, to better isolate the contribution of consistency modeling in our routing decisions.

---

### Official Review · Reviewer_fJDh · 2025-11-03

**Soundness:** 2
**Presentation:** 3
**Contribution:** 2
**Rating:** 2
**Confidence:** 3

**Summary:**

The paper introduces the Probabilistic Certainty and Consistency (PCC) framework, a novel and well-articulated approach for estimating LLM factual confidence by jointly modeling probabilistic certainty and reasoning consistency. The adaptive verification pipeline, which dynamically routes claims to different strategies based on PCC signals, is clear, rigorous, and impactful contribution. The key strength of the method is the empirical evidence that shows PCC's superior calibration (lower ECE) over existing baselines across multiple LLM families.

**Strengths:**

1. The notion of Probabilistic Certainty and Consistency (PCC) framework is conceptually and methodologically sound, and empirically shown to be superior to single-signal methods.

2. The adaptive verification pipeline, which dynamically routes claims to different strategies based on the two-dimensional PCC confidence profile, is logical for optimizing efficiency and resources.

**Weaknesses:**

**Novelty**: The key components of PCC are log-probability based certainty and NLI-based consistency, both of which are well-established uncertainty estimation methods. The novelty is mainly in how they are integrated and adapted; no fundamentally new theoretical concept or learning framework has been proposed. Therefore, the overall novelty appears to be quite light.

**Fixed threshold for routing decision**: Although the authors articulated this issue as a limitation in the Appendix, in my opinion, this is closely related to the novelty aspect of the paper as mentioned above. Thus, I recommend that the authors conduct and include a sensitivity analysis to show how performance metrics are affected by variations in $\alpha$ and $\beta$, to inform future implementations about the robustness of these thresholds.

**More recent datasets**: It is not clear why the authors did not consider datasets such as FEVEROUS, EX-FEVER, AVeriTeC, which are more recent than HOVER.

**Questions:**

Is there any reason for not considering more recent datasets, such as FEVEROUS, EX-FEVER, and AVeriTeC?

**Details Of Ethics Concerns:**

See weaknesses

---

> ### Author Response · Authors · 2025-11-21
>
> We thank the reviewer for the thoughtful comment and constructive feedback. We greatly appreciate the time and effort you have invested in reviewing our work. Below, we address each of your concerns in detail.
>
> **Q1. Novelty appears to be quite light.**
>
> Thank you for this comment. Our novelty lies not in introducing entirely new uncertainty primitives, but in how we jointly model probabilistic certainty and counterfactual reasoning consistency and use their interaction to guide verification strategies. The key contribution of PCC is the two-dimensional confidence profile, which enables adaptive routing across direct answering, targeted search, and deep search. This integration is what yields the improved calibration and accuracy demonstrated across multiple datasets and LLM families.
>
>
> **Q2. Fixed threshold for routing decision.**
>
> Thank you for this suggestion. While the thresholds are simple, they are not the core novelty of PCC. Their purpose is to operationalize the two-dimensional certainty and consistency space for routing decisions. As recommended, we are working on including a sensitivity analysis in the next version to show how performance varies under different certainty and consistency cutoffs, and to demonstrate the robustness of our routing design.
>
>
> **Q3. More recent datasets.**
>
> Thank you for raising this point. Our dataset selection is based on challenge type, rather than dataset release date. As described in Section C.1:
>  - SciFact requires specialized biomedical domain knowledge.
>  - FeLMWk consists of LLM-generated hallucinated claims, testing robustness to model-induced errors.
>  - HoVER contains multi-hop claims that require multi-step reasoning.

---

> > ### Comment · Reviewer_fJDh · 2025-11-27
> >
> > **Re-Q1.**
> > >Our novelty lies ... in how we jointly model probabilistic certainty and counterfactual reasoning consistency and use their interaction to guide verification strategies.
> >
> > >The key contribution of PCC is the two-dimensional confidence profile, which enables adaptive routing across direct answering, targeted search, and deep search."
> >
> > The paper uses a 2D confidence profile for this joint modeling. Are you claiming that such a simple profile-driven routing is a fundamentally novel contribution?
> >
> >
> > **Re-Q3.** Your response does not clearly address the reasons behind not including these benchmarks: FEVEROUS, EX-FEVER, AVeriTeC. Are they not challenging enough? Why not?

---

### Official Review · Reviewer_ARwc · 2025-11-03

**Soundness:** 3
**Presentation:** 3
**Contribution:** 3
**Rating:** 6
**Confidence:** 4

**Summary:**

This paper seeks to avoid systematic RAG usage by measuring a model's confidence in a claim in two ways. They check output token probability, as well as answer consistence when prompted for counterfactuals. They empirically demonstrate a well performing pipeline.

**Strengths:**

concrete approach with direct and obvious technical applications w.r.t. today's customer facing industrial models. The applied methods seem to be designed to be robust to a model's output variation, (use of NLI, and mapping of multiple tokens to TRUE/FALSE outputs). Reproducibility efforts are made, with prompts being provided, and good method descriptions. Extensive and relevant model and dataset use. Comparison against multiple baselines.

**Weaknesses:**

the specific problem driven pipeline makes the authors choose strong assumptions:

> Lexical uncertainty v.s. Semantic uncertainty is not directly adressed. As discussed by Kuhn et al 2023, the model could output multiple different (inconsistent) tokens that would nonetheless have the same meaning. Is there a reason why your 2 metrics would not fall prey to this?

> In this specific context of optimising RAG retrieval, I notice multiple runs are required, with extra sampling steps. It would be relevant to have an estimation of when it becomes interesting to pay this cost compared to other existing metrics. Is it systematically cheaper to sample counterfactuals from the model than it is to run retrieval?

> (Minor: maybe mention earlier that you use multiple baselines--which is a strength. early on it seems you use only verbalized confidence)

> line 172/173 makes the assumption a model "should" not easily be swayed. Why this seems somewhat instinctive form an anthropomorphic perspective, empirical results are quite strong, there is no explanation or theoretical modelisation of why this should be, and this is not tested empirically either. This seems perhaps to be more of a hypothesis.

**Questions:**

See weaknesses

---

> ### Author Response · Authors · 2025-11-21
>
> We thank the reviewer for the thoughtful comment and constructive feedback. We greatly appreciate the time and effort you have invested in reviewing our work. Below, we address each of your concerns in detail.
>
> **Q1. Lexical uncertainty vs. semantic uncertainty is not directly addressed.**
>
> We appreciate this insightful question. The task in Kuhn et al. (2023) focuses on NLG, where the model generates sentences such as “France’s capital is Paris” vs. “Paris is France’s capital.” In that setting, token-level uncertainty does not necessarily reflect semantic uncertainty. In our task, however, we constrain the model to produce only a single token that represents its classification of the claim (True vs. False). In this context, lexical uncertainty and semantic uncertainty coincide, as both are reflected in that single token. To handle LLM tokenization artifacts, we also include a normalization step that maps variants such as “.True” to “True.”
>
>
> **Q2. Concerns about extra API calls. Is it systematically cheaper to sample counterfactuals from the model than it is to run retrieval?**
>
> Thank you for this suggestion. We will include a cost analysis comparing targeted search and deep search. Sampling counterfactuals from the model is generally cheaper than calling external web search or deep search APIs. However, because fact-checking requires precision and LLMs may hallucinate, retrieval remains an essential step to ensure factual correctness. In addition, deep search provides users with a list of URLs for its information sources, which further enhances the interpretability of the pipeline.
>
>
> **Q3. Clarification on baseline selections.**
>
> Thank you for the suggestion. Our paper has two contributions: the first is the proposal of a new factual uncertainty quantification method (PCC), which we compare against verbalized confidence; the second is an end-to-end fact-checking pipeline guided by PCC, which we compare against four state-of-the-art pipelines. We will clarify this in the introduction section.
>
>
> **Q4. Assumption on a model “should” not easily be swayed under counterfactual conditions.**
>
> This assumption is indeed a hypothesis that we observe empirically, and our results show a correlation between robustness to counterfactual prompting and factual correctness. In the revised version, we will explicitly frame this as a hypothesis rather than an assumption.

---

### Note · Authors · 2026-01-04

I have read and agree with the venue's withdrawal policy on behalf of myself and my co-authors.